# Management of Social Behaviour of Domestic Yaks in Manang, Nepal: An Etho-Ethnographic Study

**DOI:** 10.3390/ani13020248

**Published:** 2023-01-10

**Authors:** Théophile Johnson, Emma Pilleboue, Maxime Herbrich, Eric Garine, Cédric Sueur

**Affiliations:** 1UMR Lesc, Université Paris Nanterre, 92000 Nanterre, France; 2IPHC, Université de Strasbourg, CNRS, UMR 7178, 67000 Strasbourg, France; 3Institut Universitaire de France, 75005 Paris, France; 4ANTHROPO-LAB, ETHICS EA 7446, Université Catholique de Lille, 59000 Lille, France

**Keywords:** anthropozoology, social network, human–animal bond, herd synchronisation, biologging

## Abstract

**Simple Summary:**

The domestic yak, a type of cattle native to the Himalayan region and Siberia, is known for its ability to withstand harsh climatic conditions at high altitudes and is primarily domesticated for its milk, wool, and meat. Most of the research on yaks has been conducted in China, where approximately 94% of the total estimated yak population can be found. In this study, the authors conducted an ethology study on the social behaviour of yaks in the Annapurna Valley of Nepal, where they are raised for their milk, wool, and meat and are also used for transportation. The authors used ethnographic inquiries to gather data on the social behaviour of yaks and the coping strategies used by herders to manage them. They also equipped cattle with one Actigraph wgt3x-BT to measure activity using an accelerometer and spatial associations using a proximity recorder. They found that yaks in both herds exhibited cohesive and synchronized behaviour, with similar activity patterns during the day and a peak of activity at night. They also observed that yaks in the first herd were more reactive to the presence of humans, while those in the second herd were more reactive to the presence of other yaks. The authors suggest that these differences in behaviour may be due to the different herding management practices used in each herd.

**Abstract:**

Herdsmen use different techniques, as per varying geographies and cultures, to keep the cohesion within herds and avoid animals getting lost or predated. However, there is no study on the social behaviour of yaks and herdsmen management practices. Therefore, this ethology study was initiated by ethnographic inquiries. In Manang, the success of the shepherd is dictated by his personal attribute of ‘Khula man’ or open-heartedness. This attribute refers to good intentions and emotions such as empathy, which allow the shepherd to focus more on others than on himself. This cultural way of assessing the skills required to become a successful and knowledgeable shepherd guided us to study the effect of cultural values on the herd’s social behaviour. We collected data from two herds living at the same settlement (Yak kharka, 4100 m altitude, Nepal) by equipping them with loggers. One of the herdsmen used the tether rope while the other one did not. Moreover, the Thaku herd had a more proactive shepherd than the Phurba one. In each herd, 17 animals were equipped with one Actigraph wgt3x-BT to measure activity using an accelerometer and spatial associations using a proximity recorder. One of the herds was equipped with GPS (N = 11) as well. Using GPS locations and activity, we showed that the two herds were cohesive and synchronised their activities but the Thaku herd (tether rope herd) was more cohesive than the Phurba herd based on the Actigraph signals. The shepherds also have personal knowledge of the social relationships of individual animals in their herds and use these relationships to keep the group cohesive and to manage cattle well.

## 1. Introduction

The domestic yak (*Bos grunniens*), a descendent of wild yak (*Bos mutus*), is domesticated in the Himalayan region and in Siberia for its milk, wool, and meat but mainly for its tolerance to harsh climatic conditions at high altitudes [1]. Central Asia and yak domestication is also known as a lactase gene locus of apparitions [2]. Approximately 94% (13.3 million) of the estimated total yak populations are found in China (Food and Agriculture Organization of the United Nations, 2003); therefore, most of the studies on yak’s grazing behaviour, livestock management, and ecological systems were conducted in China [3,4,5]. Few studies were also undertaken on its social behaviour because it is important not only as fundamental knowledge but also as practical. Herd management is crucial, particularly in places where there is an increase in the number of predator incidents. As in most cattle species, yaks also live in herds mostly comprising females and juveniles [6]. There is a sexual segregation, where males leave the herd in adulthood, and revisit only in the mating season. Himalayan inhabitants kill these isolated males for their meat. Contrastingly, female yaks are protected for producing milk, wool, and calves [7,8]. As the herds can be large, with 50–60 animals, owners employ herdsmen to keep watch on the herd and milk them. Maintaining such big herds (Figure 1a) is unnatural because wild cattle live in a smaller group size of 20; however, in plains, the number can be over 200 individuals/group [6]. Therefore, the herdsmen had to employ different techniques, varying as per geographies and cultures, to keep the herd cohesive and avoid animals from being lost or predated (Figure 1b).

Manang is a village situated at an altitude of 3500 m in the Annapurna Valley, Nepal [9]. Its inhabitants earn (make their living) by practising agriculture, rearing livestock, and trading with the tourists [10]. Due to the change of seasons, the herdsmen practise transhumance in search of greener pastures where cattle graze freely but are brought back to the camp due to the fear of predators. This is rarely combined in winter with some corals for the juveniles, but more usually with a system called tether rope. During the night, the calves are tied close to their mothers to promote cohesiveness at the camp. Interestingly, the herdsmen also choose to tie the calves together and keep the mothers close to each other to keep the herds cohesive [11]. However, until now there is no study on the social behaviour of yaks and the coping strategies that herdsmen use. This ethology study was initiated by ethnography inquiries; without these inquiries, it is difficult to formulate a hypothesis on the impact of herders on the social behaviour of the yaks.

## 2. Ethnographic Study

The shepherds’ discourses about their herds are always focused on the relations they have with each individual yak. Irrespective of the shepherd’s culture and geography he comes from, the source of knowledge comes from sharing lives. In Manang, the success of the shepherd is dictated by his personal attribute of ‘Khula man’ or open-heartedness. This attribute refers to good intentions and emotions such as empathy, which allow the shepherd to focus more on others than on himself. This cultural way of assessing the skills required to become a successful and knowledgeable shepherd guided us to study the effect of cultural values on the herd’s social behaviour.

This study was conducted over a period of 10 months, beginning in 2017 and ending in 2018. For 10 months, we observed that by strategising facilitating habits or daily rites, the shepherds nurtured the cauri (female yak) to form a good herd. Shepherds emphasised on the quality of the relationship they had with every yak in the herd as it was crucial for daily milking. The quality of the relationship was tested over milking as milking was interpreted as the cooperation of the cauri that ‘gives her milk’ to the shepherd. To achieve this, the shepherd tied all the young ones to the tether rope in particular spots, irrespective of the season. The calves of the same age were grouped together to ‘make friends’ (as herders literally say). Until three years of age, the calves were tied to the tether rope as they nursed on their mums, and the older ones took their places by themselves. The placement of each calf was chosen to prevent the juveniles and the mother yaks from fighting with each other. This helped the shepherd to find his herd in the morning, and to have a harmonious relationship at the milking station by reducing stress, improving the milk quality, and building cohesiveness amongst his herd. If not conducted this way, the calves would destroy the tether rope during the night and it would become impossible to get the herd back from the pastures, as they would not stay together or they would refuse to come back to the settlement and not cooperate. The daily movements of the herds in the pastures are implicitly related to the shepherd’s work and personality. If the shepherd is considerate, takes care of his cattle by giving *tsangba* or *kho* (a mixture prepared with salt and barley flour), and does not take punitive action, the herd stays around and comes back to the settlement. These shepherding skills impact herd behaviour and it is important to study the effect of human actions shaping the social behaviour in yaks.

The shepherds also have personal knowledge of the social relationships within their herds. They do not automatically assimilate a mother yak’s personality to her calf. The calves develop their own personalities with time. Friendships between the calves were frequent and beyond the shepherd’s control, e.g., two calves were too slow and always stayed together at the end of the herd, next to the shepherd. As frequent loneliness is a criterion of personal complexity or non-subjection and a sign of intelligence, yaks may need the cohesiveness that can be improved by the shepherds.

In the winter of 2022, the data were collected from two herds living at the same settlement (Yak kharka, 4100 m altitude) by equipping them with loggers. One herdsman used the tether rope in his herd (named Thaku) whilst in the other herd (named Phurba), a rope was not used. Moreover, the Thaku herd’s shepherd was more proactive than the Phurba one. In each herd, 17 yaks were tagged with an Actigraph wgt3x-BT to measure activity using an accelerometer and spatial associations using proximity recorders [12]. The proximity recorders helped us in measuring the cohesiveness of the herd during the daytime and to check the effect of the tether rope on spatial associations amongst the cattle. One herd was equipped with GPS (N = 11) (Figure 1c,d) as well, which helped in assessing the cohesiveness and synchronicity of motion and behaviours of the yaks.

The shepherds were excited and participated actively in fitting the loggers. They were curious at the possibility of observing their herd onscreen and seeing if the cauri were staying together or dispersing on the pasture. The equipped cattle were chosen by the shepherds, who categorised the animals in groups such as mothers and calves, good or bad ones, or two inseparable friends. The calves could not be equipped with the GPS as the loggers were heavy and some cauris could not be equipped because they were uncooperative. As a result, only the cauri that were closest to the shepherd were equipped.

## 3. Materials and Methods

### 3.1. Ethological Study Methods

#### 3.1.1. Study Subjects

Two herds were shepherded by two different herdsmen at Yak kharka, Manang, Annapurna Valley, Nepal (28.722992730245295, 83.97359502534931). During the study, at least four snow leopards were observed in the valley (from 10 to 24 March 2022). Moreover, a month before and after the study, four yaks were killed by the predator. A dog was used to keep the snow leopard away from the village. At night, the dog stayed in the village and during the day it could roam between the village and the pastures. In each herd, 17 yaks were equipped with Actigraphs [12,13] but GPS tagging was conducted on 11 yaks of the Phurba herd only. The herdsmen along with the dog were equipped with a GPS tracker and an Actigraph not only to measure their activity, spatial associations, and locations with animals but also to act as a control or reference point for the yaks. As the shepherds failed to wear the loggers all the time, the data were insufficient to analyse the relationships between the shepherds and their yaks. The two selected herds were not linked to each other and were not grazing on the same pasture, most of the time. In each herd, the 17 equipped yaks were chosen to balance the sex and age variables. However, revisiting male adults were excluded from tagging as they were too aggressive. Similarly, some female yaks were also excluded for being aggressive, though they could have been good candidates to study. In the Phurba herd, Actigraphs were fitted to eight males ageing from one to three years and nine females aged from five to thirteen years. Additionally, GPS tags were fitted on eleven cattle (two males and nine females). Similarly, in the Thaku herd, four males aged one to two years and thirteen females aged one to eleven years were fitted with Actigraphs. For reasons beyond the study’s control, only male juveniles survived the 2022 winter. During the study, the Phurba herd did not use the tether rope but the Thaku group used it from 21 March after three days of logging. The animals in the two herds left the village each morning around 6 am and were brought back to yak Kharka at approximately 5 p.m.

#### 3.1.2. Data Collection

All loggers of yaks were synchronised and every herd started at the same time.

*Activity*: Activity was measured using an accelerometer with the Actigraph wGT3X-BT [12,13]. Accelerometer sensors measure the change in speed (velocity) per unit of time (M/s^2^). To measure and assess the behaviour of animals, we did not calibrate the three axes of the accelerometer and used one axis to measure the global activity, as the measures on the three axes had a high correlation coefficient (R^2^ > 0.7). The frequency was set at 0.1 Hz (one measure every ten seconds). Owing to some battery issues, the activity was measured for seven days for the Phurba herd and for five days for the Thaku herd.

*Spatial associations*: Proximities between individuals were measured using Bluetooth technology with the Actigraph wGT3X-BT. A received signal strength indicator (RSSI) [14,15] was recorded between two identified loggers every minute (60 Hz). The signal between two loggers was strong at <1 m (−45 RSSI), became weak at 2–3 m (−90 RSSI), and recording stopped when the distance was >3 m, indicating that the Actigraphs do not work with greater distance. Moreover, a shorter distance fulfilled our purpose because studies usually consider a distance of <3 m between cattle to have positive or affiliated relationships. Due to a difference in the battery life, the absolute frequencies of the RSSI were corrected by acknowledging the minimum recorded time for a pair of yaks, which provided a relative frequency of proximities for a pair named as spatial relationship.

Matrices of spatial associations were available every day and every night for the duration of the study.

*GPS locations*: The GPS locations of the 12 Phurba herd yaks were scored from March 10 to 24, every minute during the day and every ten minutes during the night. With the help of *Dplyr* [16], *lubridate* [17], and *moveHMM* [18] R packages, the aberrant GPS points were eliminated by applying a speed threshold of 6 km/h between two locations. To obtain the trajectories with regular time intervals of 2 min between two GPS locations, the R package *AdehabitatLT* [19] was used with the functions *SetNa* and *Sett0*. Animated graphs with maps as the background were obtained by using the R package *ggplot2* [20], followed by *gganimate* [21]. Two thousand gif pictures per day were chosen for a graph. The map was added to the figure using the package *GGmap* [22] with API Maps Static (the Maps Static API Service creates a map based on URL parameters sent through a standard HTTP request and returns the map as an image). This allowed us to analyse the movements and cohesion of the yaks. All these animated gif images are available on Zenodo (https://doi.org/10.5281/zenodo.7281214, accessed on 4 November 2022).

#### 3.1.3. Data Analyses

*Activity*: Kruskal–Wallis tests were used to test the difference in the activity distribution between the two herds and between juveniles and adults in the two herds.

*Social Network Analyses*: Firstly, we calculated the mean number of signals exchanged between loggers during the day and at night for each group and for each cattle (as indicated above, this mean number was corrected according to the time of the recording). Secondly, the spatial association matrices were uploaded on Gephi 0.92 [23], which were transformed into social networks. From these networks, we calculated the maximum modularity [24], which measured the strength of division of a network into subgroups. We analysed the effect of groups, their nictemeral rhythm, and the presence of the tether rope using Wilcoxon tests based on the mean number of exchanged signals and the modularity. However, as the modularity was rarely superior to 0.3, indicating that groups could not be further divided into communities, we did not assess which animal belonged to each community. We used a Spearman correlation test to correlate the mean received number of signals per individual with age.

*Spatial cohesion*: Using the GPS locations of yaks, after every 30 m we analysed the number of subgroups. We defined a cluster comprised of one or several cohesive individuals (less than 0.5° longitude or latitude of separation) separated from other individuals (or subgroups) by at least 0.5° longitude or latitude.

All analyses were made using RStudio 2022.02.2 [25], α = 0.05. We summed the values for the two groups for some tests and indicated the statistical differences between the two groups. Data are available at Zenodo: https://doi.org/10.5281/zenodo.7281214, accessed on 4 November 2022.

## 4. Results

*Activity*: The activity distribution was the same between the two herds (*p* = 0.92, Figure 2A,B). Moreover, the activity profile was not different between adults and juveniles within the Phurba herd (*p* = 0.16, Figure 1c), and Thaku herd (*p* = 0.17, Figure 1d). At 5 a.m., the activity increased as the animals left the camp. Grazing was classified as a resting period and activity was observed again at 5 p.m. when the yaks returned to the camp. The activity was lowest at night, but interestingly around midnight the activity increased except in the Thaku herd juveniles because they were attached to the tether rope.

*Social networks*: More Actigraph signals per individual were exchanged (W = 54, *p* = 0.02) in the Thaku group (mean = 77.5 ± 38.4), indicating that the cattle in the Thaku group were more cohesive than the Phurba (mean = 66.4 ± 32.2) herd. The groups were more cohesive during the night than during the day (W = 256, *p* < 0.0001, mean_night_ = 67.8 ± 30.8, mean_day_ = 37.6 ± 40.3, Figure 1) and the cohesion was higher in the presence of the tether rope (W = 3, *p* = 0.025, mean_with_ = 108 ± 17, mean_without_ = 34 ± 19). In the Thaku group, more signals were received in young individuals due to the tether rope (W = 0, *p* = 0.001, Figure 3e). Contrastingly, the modularity was the same between the two groups (W = 36, *p* = 0.713, mean_thaku_ = 0.26 ± 0.09, mean_phurba_ = 0.24 ± 0.08); groups were more clustered at night than during the day (W = 54, *p* = 0.02, mean_night_ = 0.26 ± 0.09, mean_day_ = 0.24 ± 0.08) and the tethering rope increased the modularity (W = 3, *p* = 0.03, mean_with_ = 0.35 ± 0.08, mean_without_ = 0.23 ± 0.07). The modularity was significant in the case of the tether rope (Q > 0.03) as the young individuals clustered together. Finally, the mean number of signals received per individual negatively correlated with age, with younger members being closer to other members than old females (r = −0.52, S = 8343, *p* = 0.002).

*Spatial cohesion*: The mean number of subgroups observed using GPS was 1.22 ± 0.46. The herd was cohesive (one group) in 78.7% of the scans and two subgroups were observed in 20% of the cases. Three and four subgroups were observed in 1.2% and 0.1% of the scans, respectively. The 24 h period analysis suggested that cohesion varies temporally (Kruskal–Wallis test: χ^2^ = 47, df = 23, *p* = 0.002) but the pairwise comparison tests did not reveal a difference between each hour (*p* > 0.1). So, a day was divided into six periods of four hours each and the degree of cohesiveness differed during the periods (Kruskal–Wallis test: χ^2^ = 29, df = 5, *p* < 0.0001). Higher cohesiveness was observed between 1 a.m. and 4 a.m. than other periods (*p* < 0.002) except for the period between 5 a.m. and 8 a.m. (*p* = 0.09). The mean number of clusters between 1 a.m. and 4 a.m. were 1.05 ± 0.23 whilst in other periods the number was around 1.26 ± 0.58. A lower spatial cohesion was observed around 5 p.m. when yaks returned to the village (1.34 ± 0.58).

## 5. Discussion

This study is preliminary to deeper research required on the social behaviour of yaks and impact of domestication on this behaviour. This is the first study to highlight the social behaviour in yaks.

First, whilst the herds had different herding management, animals in the two herds were cohesive and synchronised their activities. Though the herds were rarely seen on the same pasture, their activity distribution was the same during the day. This distribution was punctuated by the phases of leaving the village (the yaks were doing this on their own, which is well known in domestic and wild bovines as collective moves or activity synchronisation [26,27,28,29]) and the phases of returning to the settlement, led by the herdsman. Interestingly, there is a peak of activity during the night and its reoccurrence suggested that it was not due to the presence of predators but more because of yaks going to graze in vicinity of the village. Environmental conditions such as the cold may dictate the animals’ night activity such as grazing or mating [30,31,32], but it was not the mating season and we do not believe that the yaks were cold (the temperature at night was about minus 10 °C) [33,34]. However, calves may want to nurse, which can explain the activity peak. Herd synchronisation was confirmed by the spatial cohesion of the Phurba herd. Eleven animals were fitted with GPS; the animals were chosen according to their age and personalities, where few animals were identified as independent or not social by the herdsman. The results show that 80% of the time the herd was cohesive, especially at night. So, the results indicate that with different herd management practices, domestic yaks stayed cohesive in small groups of females with juveniles and did not live as wild yaks or as a wild bovine species [1,33,35]. However, we noticed that some individuals in both the herds were independent and at four years of age, the cohesiveness with the herd reduced in males. This observation needs confirmation even though it seems coherent in the pastoral system that males are more independent and live separately from the herds. It would be interesting to assess if this cohesion was due to genetic selection during domestication or behavioural that can be lost. We also observed that in the nearby valley the yaks were kept feral by the villagers. It was interesting to study their behaviour further, but it was a challenge to fit these animals with loggers.

When observing individual cohesion, young yaks were more cohesive and exchanged more Actigraph signals amongst them and with mature females, but less with the older ones. Usually, in bovids, wild or domesticated, old individuals are more socially central than younger ones (e.g., Highland cattle [36]) or both have similar social centralities (e.g., European bison [35]). Younger ones get their centrality not only by sharing strong links with their mother, but also by forming a crèche (or nursery) and staying together [35]. This was observed in the Phurba group, but based on the Actigraph signal results, another force ruled the Thaku group, which used the tether rope. In the Thaku group, young individuals were very cohesive and clustered at night, but the effect seems to be prolonged during the day as the Thaku group was more cohesive during the day than the Phurba group. We may, however, wonder if the tether rope may have negative effects on the young individuals as they are unable to defend themselves or to escape from predators. Nevertheless, the herdsmen never reported attacks on juveniles attached to the rope as this system is close to, and even between, human houses. Moreover, the dog stayed in the village and in a way protects the cattle. So, the tether rope seems to be a good tool for the short-term (at night) and long-term maintenance of cohesion and reinforcement of relationships of yaks.

## 6. Conclusions

This study had two limitations: (a) the duration of the study and (b) the number of cattle equipped with loggers. Technological advances can mitigate the constraints met at this altitude that affected data scoring (the extreme cold and lack of electricity affected the batteries). However, this study shows the importance of herd management on yaks’ social behaviour, particularly with the resurgence of the snow leopard [37,38] and climate changes that can dramatically modify the relationships between the predators and wild and domesticated ungulates [39,40]. These results suggest that more studies are needed to understand how herdsmen use social relationships in yaks facing these environmental pressures [41].

## Figures and Tables

**Figure 1 animals-13-00248-f001:**
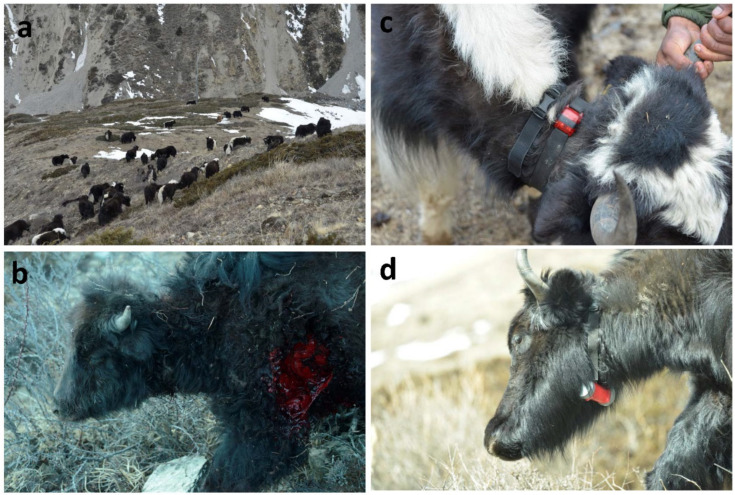
Domestic yaks. (**a**) Herd of yaks. (**b**) Juvenile yak injured by a snow leopard. (**c**,**d**) Yaks tagged with an Actigraph and a GPS to measure activity, proximity, and locations.

**Figure 2 animals-13-00248-f002:**
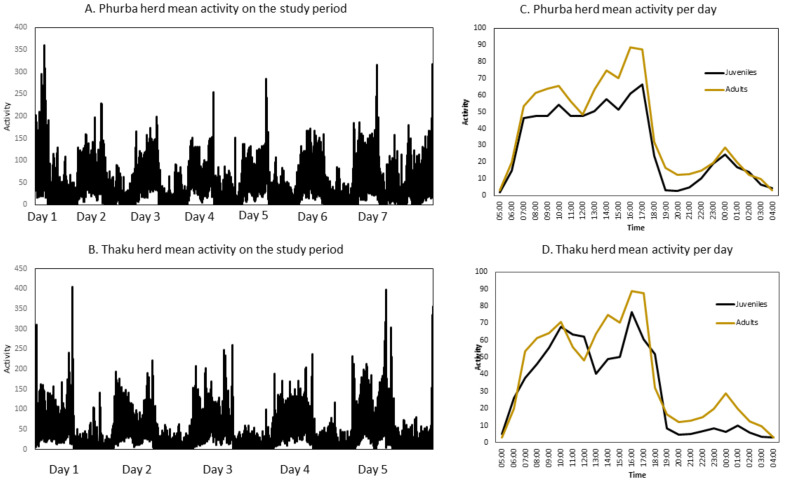
Activity rate of herd yaks measured using Actigraph.

**Figure 3 animals-13-00248-f003:**
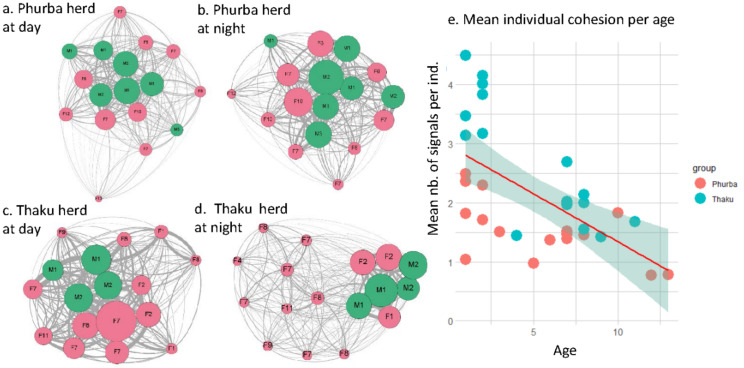
Social networks of Thaku and Phurba herds during the day and at night (from (**a**–**d**)). (**e**) Mean number of Actigraph signals/individual according to age. Pink refers to females and green to males.

## Data Availability

Data are available at Zenodo (https://doi.org/10.5281/zenodo.7281214, accessed on 4 November 2022).

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
