# Peer review of "Management of Social Behaviour of Domestic Yaks in Manang, Nepal: An Etho-Ethnographic Study"

_animals, 2023, doi:10.3390/ani13020248_

Round 1

Author Response

Q: Thank you to the authors, I enjoyed reading this article. See detailed comments:

A: Thank you for your nice comments!

Q: Row 12 Delete “on” before herdsmen.

A: Done

Q: Row 15 Change ownself with himself

A: Done

Q: Row 16 Knowledgable change in knowledgeable

A: Done

Q: Row 17 Behavior change behaviour

A: Done

Q: Row 19 Change herdsman with herdsmen

A: Done

Q: Row 21 Add “an” before accelerometer

A: Done

Q: Row 36 Populations

A: Done

Q: Row 41 Predators

A: Done

Q: Row 46 Change “It” with “them”

A: Done

Q: Row 60-61 Check the sentence

A: Done

Q: Row 65 Behavior change behaviour

A: Done

Q: Row 74 check as row 15

A: Done

Q: Row 75 check as row 16

A: Done

Q: Row 76 check as row 17

A: Done

Q: Row 82 Achieve

A: Done

Q: Row 83 delete “a” before particular

A: Done

Q: Row 89 Improing and bulding

A: Done

Q: Row 96 delte “the” before herd

A: Done

Q: Row 103 non-subjaction change with non-subjection

A: Done

Q: Row 110 Add “s” in recorder

A: Done

Q: Row 111 Checking

A: Done

Q: Row 116 Seeing

A: Done

Q: Row 133 Change ware in were

A: Done

Q: Row 136 Change euipped in equipped

A: Done

Q: Row 142 Missing space between each and other

A: Done

Q: Row 146 Aging with Ageing

A: Done

Q: Row 147 Aging in Aged

A: Done

Q: Row 157 Add An before Accelerometer

A: Done

Q: Row 158 Add of between unit and time

A: Done

Q: Row 162 Change cofficient in coefficient

A: Done

Q: Row 167 Beame with became

A: Done

Q: Row 170 Cattle (Delete S near all cattle)

A: Done

Q: Row 176 delete for

A: Done

Q: Row 214 Add “the ”before same

A: Done

Reviewer 2 Report

Thank you for the opportunity to review this paper. 

You do need to check your Figure 3.d. caption - I think it needs to say "Thaku herd at night"

Overall, this is an interesting beginning look at how herds move together through their local geography. As both herds moved together to pasture and returned to the village, I'm not sure I'm convinced that the tether rope provides a substantive difference in cohesion, even if there is statistical significance in how closely the individuals within the herd moved with the group. You do indicate that the Thaku herd (tether-rope herd) was "more cohesive" than the Phurba herd based on the actigraph signals.

Rather, I wonder if the use of the tether rope has any negative effects as younger individuals, conceivably less able to defend themselves, are tied together and their ability to make individual choices is limited. There doesn't seem to be any attempt to discuss the drawbacks of the tether rope - although I'm not sure the authors wanted to do so as no recommendations for or against its use was observed in the paper. Thus, I'm not sure if you intend to make a case for or against the use of the tether rope based on this information. The agency and subjectivity of the yaks, especially in the group using the tether rope, seems to be lacking attention in the discussion. 

Overall, it is an interesting look at herd dynamics within two different herds of yaks who differ in how they are managed by their shepherds via the use or absence of the tether rope. Authors indicate this is a preliminary study and it reads as such. The limitations and future directions statements are appreciated and seem to be sufficient for a preliminary study.

Thank you for pursuing this interesting first look at yak social behavior within a herd.

Author Response

Q: Thank you for the opportunity to review this paper. 

A: Thank you for your nice comments!

Q: You do need to check your Figure 3.d. caption - I think it needs to say "Thaku herd at night"

A: Corrected

Q: Overall, this is an interesting beginning look at how herds move together through their local geography. As both herds moved together to pasture and returned to the village, I'm not sure I'm convinced that the tether rope provides a substantive difference in cohesion, even if there is statistical significance in how closely the individuals within the herd moved with the group. You do indicate that the Thaku herd (tether-rope herd) was "more cohesive" than the Phurba herd based on the actigraph signals.

A: Done, we changed it in the abstract and in the discussion.

Q: Rather, I wonder if the use of the tether rope has any negative effects as younger individuals, conceivably less able to defend themselves, are tied together and their ability to make individual choices is limited. There doesn't seem to be any attempt to discuss the drawbacks of the tether rope - although I'm not sure the authors wanted to do so as no recommendations for or against its use was observed in the paper. Thus, I'm not sure if you intend to make a case for or against the use of the tether rope based on this information. The agency and subjectivity of the yaks, especially in the group using the tether rope, seems to be lacking attention in the discussion. 

A: We added a paragraph about this in the discussion: “We may however wonder if the tether rope may have negative effects on the young individuals as they are unable to defend themselves and to escape from predators. Nevertheless, the herdsmen never reported attacks on juveniles attached to the rope as this system is close and even between human houses. Moreover, the dog stayed in the village and protect in a way the cattle. So the tether rope seems to be a good tool for a short-term (at night) and a long-term maintenance of cohesion of yaks.” lines 292-297.

Q: Overall, it is an interesting look at herd dynamics within two different herds of yaks who differ in how they are managed by their shepherds via the use or absence of the tether rope. Authors indicate this is a preliminary study and it reads as such. The limitations and future directions statements are appreciated and seem to be sufficient for a preliminary study.

Thank you for pursuing this interesting first look at yak social behavior within a herd.

A: Thank you again for your appreciated comments.